# Catch per Unit Effort Dynamic of Yellowfin Tuna Related to Sea Surface Temperature and Chlorophyll in Southern Indonesia

**Budy Wiryawan** [1,2,*], **Neil Loneragan** [1,2], **Ulfah Mardhiah** [3], **Sonja Kleinertz** [4,5], **Prihatin Ika Wahyuningrum** [1], **Jessica Pingkan** [3], **Wildan** [6], **Putra Satria Timur** [6], **Deirdre Duggan** [6] and **Irfan Yulianto** [1,3]

1   Department of Fishery Resources Utilization, Faculty of Fisheries and Marine Sciences,
    IPB University (Bogor Agricultural University), Bogor 16680, Indonesia;
    n.loneragan@murdoch.edu.au (N.L.); prieha@gmail.com (P.I.W.); iyulianto@wcs.org (I.Y.)
2   Environmental and Conservation Sciences, College of SHEE, Murdoch University, Perth 6150, Australia
3   Wildlife Conservation Society–Indonesia Program, Bogor 16128, Indonesia; umardhiah@wcs.org (U.M.);
    jpingkan@wcs.org (J.P.)
4   Professorship for Aquaculture and Sea-Ranching, University of Rostock, 18051 Rostock, Germany;
    sonja.kleinertz@uni-rostock.de
5   Faculty of Fisheries and Marine Sciences, IPB University (Bogor Agricultural University),
    Bogor 16680, Indonesia
6   Masyarakat dan Perikanan Indonesia Foundation, Denpasar 80223, Indonesia; wildan@mdpi.or.id (W.);
    timur@mdpi.or.id (P.S.T.); dnidubh@gmail.com (D.D.)
*   Correspondence: bud@psp-ipb.org

**Abstract:** Tuna fisheries are the most valuable fisheries in the world, with an estimated market value of at least US$42 billion in 2018. Indonesia plays an important role in the global tuna fisheries and has committed to improve its fisheries management; therefore, a pilot of long-term spatial-temporal data bases was developed in 2012, however none have utilized data to have better understanding for management improvement. In this study, the annual and seasonal variation of large (≥10 kg) Yellowfin Tuna (YFT) catch per unit effort (CPUE) have been investigated and the influence of sea surface temperature (SST) and chlorophyll-*a* on these variables examined. We used fish landing data from West Nusa Tenggara recorded every month between 2012 and 2017 and analyzed using generalized linear models and generalized additive models. We found a seasonal and annual pattern of tuna abundance affected by SST and chlorophyll-*a* (chl *a*) and related to upwelling and El Nino event. These results also suggest that a two-month closure to fishing in August and September in southern Lombok is worth considering by the Government to maximize conservation of stocks due to a high abundance of juveniles emerging during the upwelling months from June to August.

**Keywords:** Chlorophyll A; sea surface temperature; standardized catch per unit effort; *Thunnus albacares*; generalized linear model; generalized additive model; tuna fishery; relative abundance; Labuhan Lombok

## 1. Introduction

Tuna is one of the globally most important fish, caught by fishing vessels of more than 85 countries and contributing 5.7% or 4.8 million t to the global seafood production in the year 2016 [1,2]. In 2016, the value of tuna at all landing sites in the world was estimated to be more than US$10 billion annually [2], with approximately an additional US$42.2 billion market value [3]. They provide food, key nutrients, and income to coastal countries and their local communities [4]. Indonesia's tuna

fisheries are amongst the most important, diverse, and complex fisheries in the world. More than 130,000 vessels from 3 m in length to more than 100 m are operated with a total catch of more than 1 million tons of tuna per year [2,5]. Tuna is one of the most valuable species in Indonesia, with an export value of US$677.9 million in 2017. Hence, tuna is an important generator of wealth across all fleets, making it the most important seafood category for the country. The species of tuna caught in Indonesian waters include yellowfin (*Thunnus albacares*), bigeye (*Thunnus obesus*), albacore (*Thunnus alalunga*), and skipjack (*Katsuwonus pelamis*).

Tuna fishing has grown significantly in the waters of the Indonesian archipelago so that in 2016, Indonesia was the top country globally for tuna landings [2,6]. In the last 10 years, Indonesia has become involved in several regional fisheries management organizations, such as Indian Ocean Tuna Commission, Commission for the Conservation of Southern Bluefin Tuna, Western and Central Pacific Fisheries Commission, and the Inter-American Tropical Tuna Commission [7]. The Government of Indonesia has shown its commitment to sustainable tuna fisheries by issuing a Tuna Fisheries Management Plan in 2015 [7]. One of the most important components of this plan is the collection of data to improve decision support systems for tuna fisheries management. A key issue for fisheries scientists is the determination of fishing effort from the fishing fleets to be evaluated in relation to fishing mortality and it is commonly used as a basic input in fish stock assessment models [8]. Knowing yield per unit area and catch per unit effort can be complementary indicators of the stock status [9]; the first refers to the total catch without any reference to the fishing pressure, while the second one links the catch to the fishing efforts. Numerous projects and organizations have been involved in supporting the Government of Indonesia to develop data collection systems for tuna, including the Indonesia Marine and Climate Support (IMACs) project of United States Agency for International Development (USAID) Indonesia that developed the original Ifish system in 2012, which is now administered by Provincial Data Management Committees [10]. Data from tuna fishing boats are collected by enumerators through standardized methods and then uploaded to an online database (Ifish data) that is accessible to fishers, academicians and researchers, industries, and government. The Ifish data are available for advanced studies, awareness, policy development, and decision-making [10].

The abundance and composition of tuna resources caught by fishers are closely related to human factors, such as accessibility, proxy of fishing activity (boats characteristic), and experience in the sea, and other natural factors, such as weather condition, SST, chl *a* concentration, and the state of fish resources. Structural changes of the fishery resources are highly related to the utilization intensity of those resources. The oceanic features such as current dynamics, fronts, eddies, and convergences [11], and SST have been used to investigate productive frontal zones [12], which can further be used to indicate potential tuna fishing grounds. Thermal (or color) gradients in satellite images that arise from the circulation of water masses often indicate areas of high productivity [13]. Meanwhile, chl *a* data can also be used as a valuable indicator of water mass boundaries and may identify upwelling that can influence tuna distribution within a region [14,15]. Chl *a* concentrations over 0.20 mg·m$^{-3}$, indicate high phytoplankton abundance for tuna prey, such as small pelagic fishery resources [16,17].

Fisheries in West Nusa Tenggara, one of the 34 provinces in Indonesia (Figure 1), are characterized as a deep-sea handline tuna fishery. This province contributes significantly to the national production of tuna, with yellowfin tuna, *T. albacares*, the most abundant species landed, comprising ~70% of the total catch [18]. Despite the significance of tuna in this province, few studies related to tuna fisheries have been made in this region [19]. We utilize the 'Ifish database' of the Data Management Committee, West Nusa Tenggara from 2012 to 2017, to investigate how CPUE of large tuna varies among different temporal factors, such as year, month of the year, different fishing days, and boat lengths, as well as the use of fish aggregation devices. The variation in CPUE of large *T. albacares* (>10 kg) related to sea-surface temperature and chl *a* is also examined. Thousands of records of CPUE from the Ifish database are analyzed using General Additive Models (GAMs) and Generalized Linear Models (GLMs) to evaluate the importance of these difference factors. Finally, we use the selected most parsimonious model to understand the seasonal variation of tuna resources and relative abundance related to environmental

factors, i.e., chl *a* and sea surface temperature, aiming to provide recommendations for sustainable utilization and management of the tuna fishery.

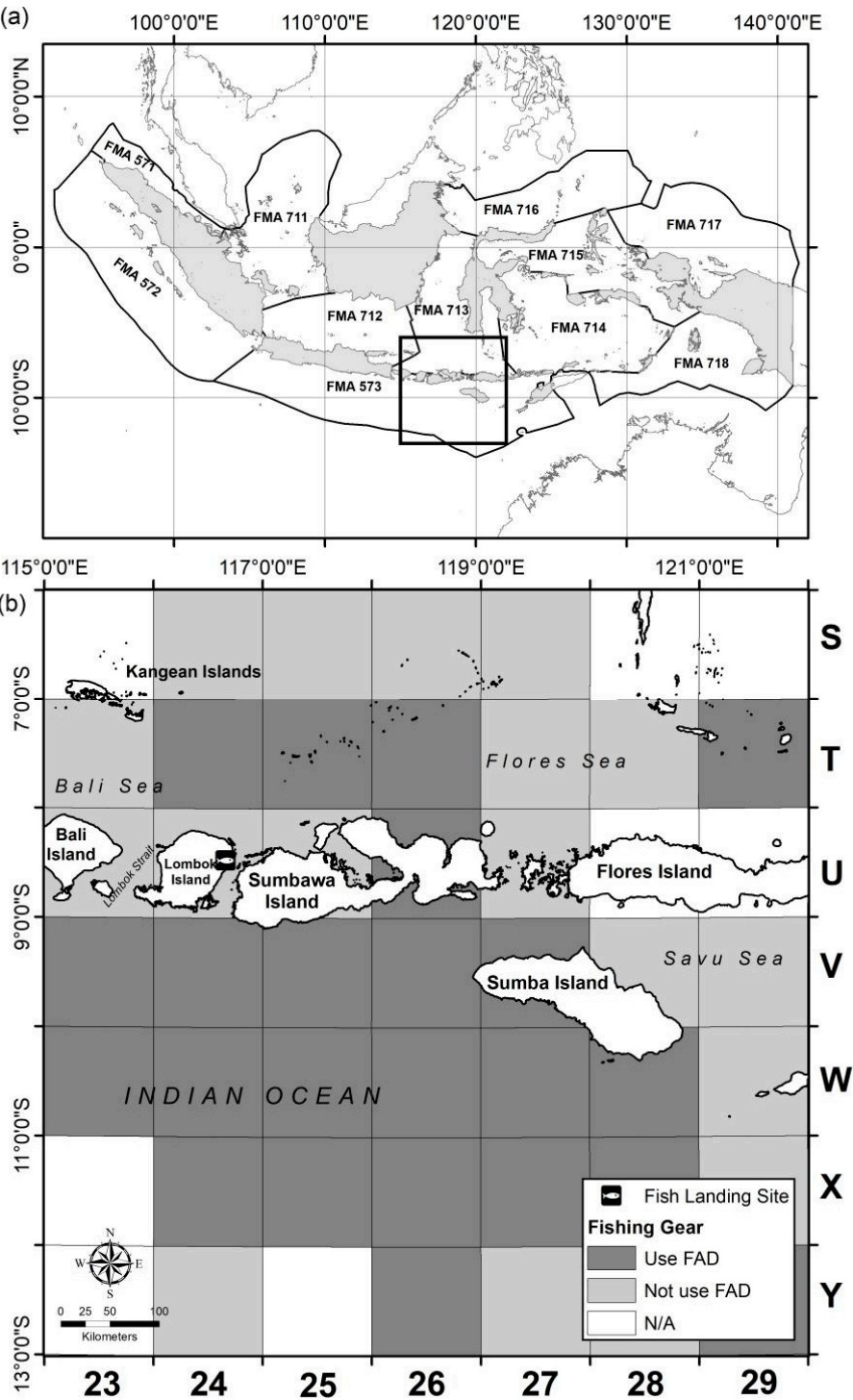

**Figure 1.** Map of (**a**) Indonesian Fisheries Management Areas (FMAs) showing the study region within West Nusa Tenggara and (**b**) fish landing and monitoring site, as well as the grids covering the fishing ground in the eastern Indian Ocean, Bali and Flores Seas, West Nusa Tenggara, Indonesia; combinations of letters (right) and numbers (below) are the grid codes for the fishing ground. N/A = not applicable as there are no records from these grids.

## 2. Results

### 2.1. Distribution and Variation of Nominal Catch per Unit Effort

In total, 2831 trips caught YFT using handlines from 2012 to 2017, with 596 of these trips (~21.1%) fishing around fish aggregating devices (FADs). A total of 41 different grids were fished during all trips, with FAD fishing taking place in 23 of these grids (Figure 1b). The average total catch of large YFT was 15 fish from fishing trips with an average duration of 11 days (Table 1). The lowest mean catch was 10 fish per trip in 2010 and the greatest mean was 25 per trip in 2012. The average catch using FAD was 13 tunas per trip, while without FAD was 15 individuals per trip. The SST and chl *a* concentration at all fishing grounds ranged from 25.7 to 32.0 °C and 0.07 to 1.80 mg·m$^{-3}$, respectively. The lowest mean annual SST was 28.3 °C in 2017, while the highest mean SST was 30.5 °C in 2012. For chl *a*, the lowest and the highest were 0.17 mg·m$^{-3}$ (2012) and 0.42 mg·m$^{-3}$ (2017), respectively (Table 1, Figures 2 and 3).

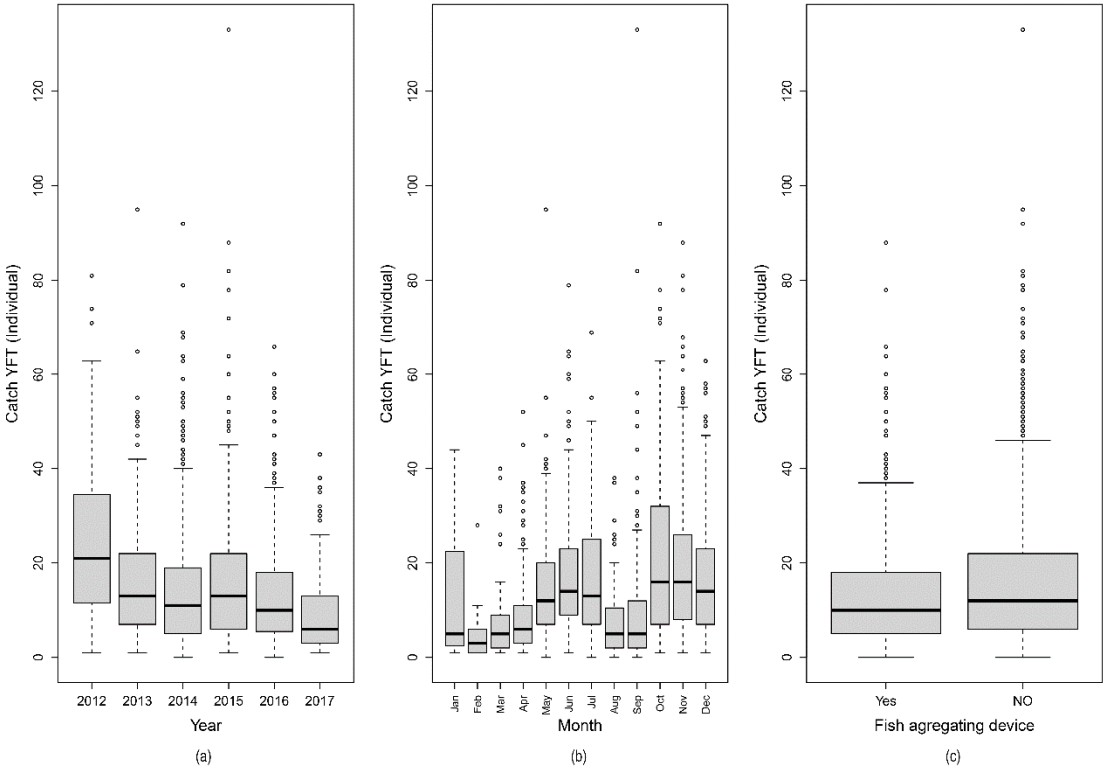

**Figure 2.** Box plots of catch per trip, number of large (>10 kg) Yellowfin Tuna (YFT), *Thunnus albacares*, YFT individuals per trip by (**a**) year, (**b**) monthly average from all years, and (**c**) fish aggregating device. Dark horizontal line shows the median, boxes cover the 25th and 75th percentiles and the whiskers show the 5th and 95th percentiles.

**Table 1.** Summary of the variables used in the analysis of large yellowfin tuna (>10 kg), *Thunnus albacares*, caught from West Nusa Tenggara, Indonesia, showing the mean value (±1 SD) for Yellowfin Tuna catch, trip duration, length of boat, SST, and chlorophyll-*a* (chl *a*) from this study period (2012–2017).

| Year | Catch of YFT (Individual) | Trip Duration (Days) | Length of Boat (m) | SST (°C) | Chl *a* (mg m$^{-3}$) |
|---|---|---|---|---|---|
| | Mean ± SD | Mean ± SD | Mean ± SD | Mean ± SD | Mean ± SD |
| 2012 (*n* = 131) | 25 ± 18 | 10.45 ± 3.11 | 17.30 ± 3.63 | 30.48 ± 0.72 | 0.17 ± 0.03 |
| 2013 (*n* = 816) | 15 ± 11 | 10.73 ± 3.60 | 16.03 ± 2.97 | 29.07 ± 1.05 | 0.24 ± 0.12 |
| 2014 (*n* = 796) | 14 ± 12 | 11.84 ± 3.69 | 14.34 ± 2.65 | 28.48 ± 1.35 | 0.33 ± 0.25 |
| 2015 (*n* = 603) | 16 ± 14 | 12.16 ± 3.36 | 14.58 ± 2.81 | 28.50 ± 1.42 | 0.33 ± 0.20 |
| 2016 (*n* = 400) | 14 ± 12 | 11.61 ± 3.73 | 13.62 ± 2.33 | 29.55 ± 1.11 | 0.18 ± 0.06 |
| 2017 (*n* = 117) | 10 ± 9 | 11.65 ± 3.77 | 13.23 ± 1.75 | 28.29 ± 1.13 | 0.42 ± 0.31 |
| 2012–2017 (*n* = 2831) | 15 ± 13 | 11.48 ± 3.63 | 14.87 ± 2.95 | 28.89 ± 1.33 | 0.28 ± 0.19 |

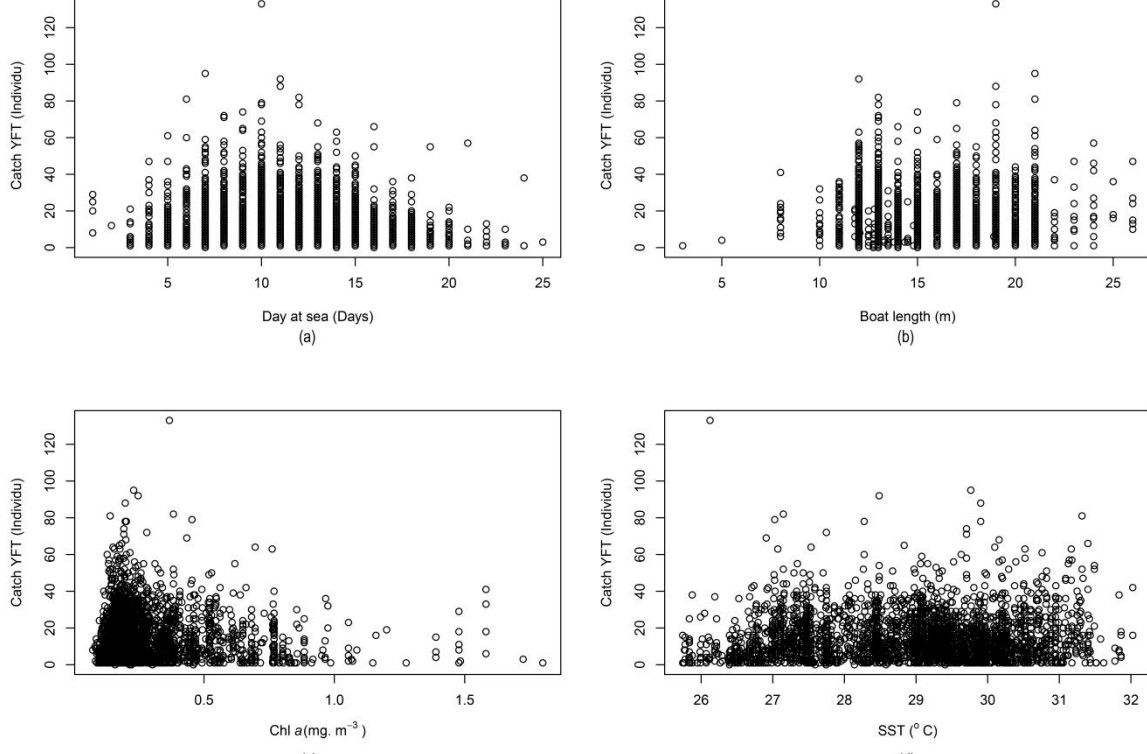

**Figure 3.** Plots of the total catches of large YFT versus (**a**) day at sea, (**b**) boat length, (**c**) chl *a*, and (**d**) sea surface temperature (SST).

An initial comparison of the total catches of large YFT (nominal/unstandardized catches) using analysis of variance indicated that they differed significantly among years and months ($p < 0.01$, Table 2). The highest and lowest annual mean catches were recorded in 2012 (25 individuals) and 2017 (10 individuals), respectively, while the highest and lowest monthly mean catches were in February (7 individuals) and October (20 individuals), respectively. Average catch of YFT (nominal/unstandardized catches) at the areas (grids) with FADs was significantly higher ($p < 0.001$, Table 2) than the catch at the areas (grids) without FAD. We also detected some nonlinear patterns when we plotted the YFT catches against the duration of fishing trips (days at sea), boat length, SST, and chl *a* concentration (Figures 2 and 3).

**Table 2.** Analysis of variance for nominal catch of large Yellowfin Tuna (>10 kg), *Thunnus albacares,* landed in West Nusa Tenggara, Indonesia.

| Parameter | Degree of Freedom | Sum of Square | Mean of Square | F Value | *p*-Value |
|---|---|---|---|---|---|
| Year | 5 | 18,740 | 3748 | 24.57 | $<2 \times 10^{-16}$ |
| Residuals | 2825 | 430,887 | 153 | | |
| Month | 11 | 38,558 | 3505 | 24.04 | $<2 \times 10^{-16}$ |
| Residuals | 2819 | 411,069 | 146 | | |
| FAD | 1 | 1788 | 1787.9 | 11.29 | 0.000788 |
| Residuals | 2829 | 447,839 | 158.3 | | |

## 2.2. Distribution and Variation of Standardized Catch per Unit Effort

When all factors were included in a single GLM and GAM, the most parsimonious model to explain the CPUE of large YFT included year, month, days at sea, boat length, chl *a*, and SST. We excluded the FAD because it did not significantly affect the standardized YFT catch. Later, we found that GLM models with higher Akaike information criterion (AIC) values had residuals that were not homogeneous (Table 3, Appendix A). Meanwhile, GAM models showed lower AIC with no obvious pattern of residuals detected (Table 3, Appendix A), hence we selected the generalized additive model to predict our data. We found that the best fitted GAM was: GAM Negative Binomial (NB) Model: CPUE~Year + month + spline (boat length) + spline (days at sea) + spline (Chl *a*) + spline (SST).

The factors of month, days at sea, year, boat length, and chl *a* concentration significantly affected the catch of YFT as a parametric effect, while days at sea, SST, and Chl *a* were significant when included as non-parametric terms (Table 4, Figure 4).

**Table 3.** Description of the different models used to test the best fit model for large Yellowfin Tuna (≥10 kg), *Thunnus albacares,* and their over-dispersion and AIC values to select the "best" model (lo = loess/local regression; span = value to control loess smoothing (0.1–0.9); s() = smoothing spline; θ = df = degrees of freedom. Plots of residuals for each model are shown in Appendix A.

| No | Full Model | Over-Dispersion | AIC |
|---|---|---|---|
| | **Generalized Linear Models** | | |
| 1 | CPUE ~ Year + month + boat length + days at sea + Chl *a* + SST; family = Gaussian | NA | 22,263.00 |
| 2 | CPUE ~ Year + month + boat length + days at sea + Chl *a* + SST; family = Poisson | 9.8 | 37,384.00 |
| 3 | CPUE ~ Year + month + boat length + days at sea + chl *a* + SST; family = negative-binomial | 9.8 | 20,789.00 |
| | **Generalized Additive Models** | | |
| 4 | CPUE~ Year + month + lo(boat length, span = 0.5) + lo(days at sea, span = 0.5) + lo(Chl *a*, span = 0.4) + lo(Chl *a*, span = 0.2); family = negative binomial (θ = 1.75) | NA | 20,415.73 |
| 5 | CPUE ~ Year + month + s(boat length, df = 3) + s(days at sea, df = 5.5) + s(Chl *a*, df = 9.4) + s(SST, df = 9); family = negative binomial (θ = 1.8) | NA | 20,403.25 |

**Table 4.** Summary for the best fitted Generalised Additive Model (GAM) for large Yellowfin Tuna (>10 kg), *Thunnus albacares,* landed in West Nusa Tenggara, Indonesia.

| Parameter | Degree of Freedom | Sum of Square | Mean of Square | % MS | F Value | *p*-Value |
|---|---|---|---|---|---|---|
| ANOVA for Parametric Effects | | | | | | |
| Month | 11 | 309.93 | 28.1752 | 35.64 | 27.88 | >0.0001 |
| Year | 5 | 65.61 | 13.1227 | 16.60 | 12.9852 | $1.63 \times 10^{-12}$ |
| Chl *a* | 1 | 5.32 | 5.3237 | 6.73 | 5.2679 | 0.022 |
| Boat length | 1 | 6.85 | 6.8485 | 8.66 | 6.7768 | 0.009 |
| SST | 1 | 0.08 | 0.0838 | 0.11 | 0.0829 | 0.773458 |
| Days at sea | 1 | 24.49 | 24.492 | 30.98 | 24.2354 | $9.02 \times 10^{-07}$ |
| Residuals | 2787 | 2816.51 | 1.0106 | 1.28 | | |
| ANOVA for Nonparametric Effects | | | | | | |
| Month | | | | | | |
| Year | | | | | | |
| Chl *a* | 8.4 | | | | 2.1923 | 0.02299 |
| Boat length | 2 | | | | 2.8834 | 0.05611 |
| SST | 8.1 | | | | 6.072 | $7.65 \times 10^{-08}$ |
| Days at sea | 4.5 | | | | 22.0429 | $<2.2 \times 10^{-16}$ |

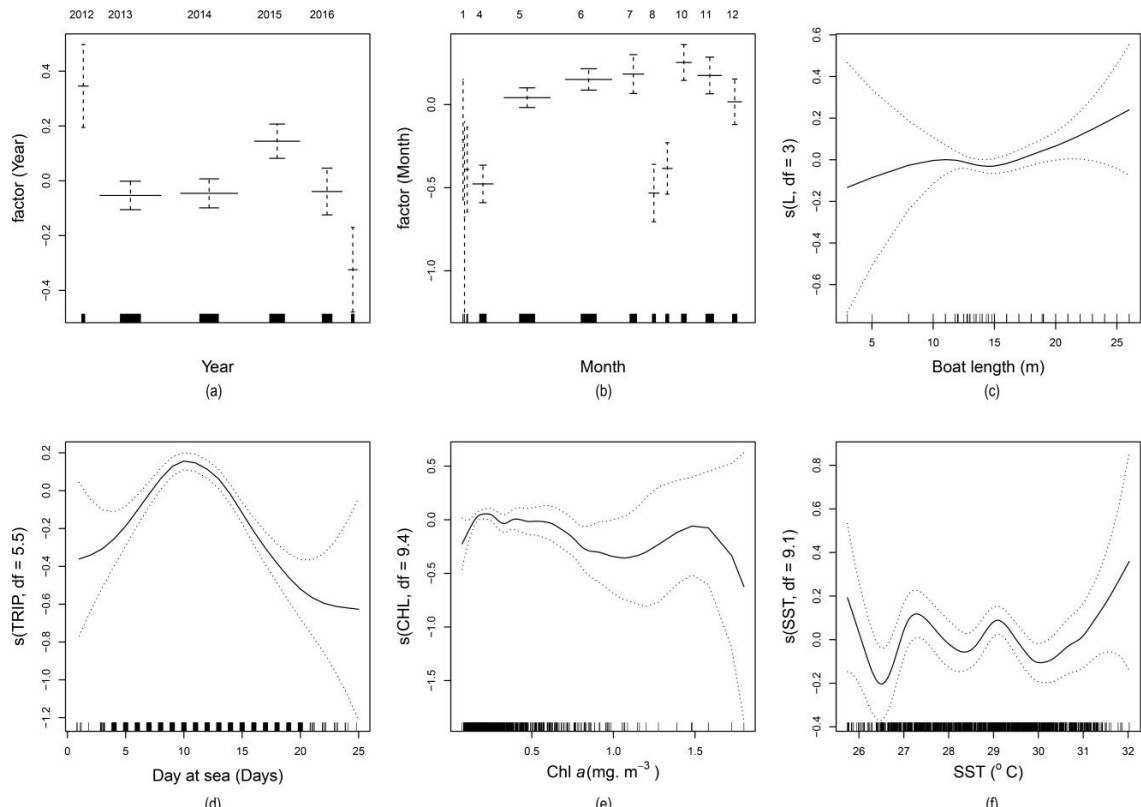

**Figure 4.** Factor (year and month) and smoothing line (boat length, days at sea, chl *a*, and SST) of generalized additive model for number of large (>10 kg) Yellowfin Tuna, *Thunnus albacares,* individuals per trip by (**a**) year, (**b**) month, (**c**) boat length, (**d**) days at sea, (**e**) chl *a*, and (**f**) SST; degrees of freedom were determined using cross validation. Density of dots on X-axis reflects the number of points.

The standardized CPUE according to the estimated marginal mean of YFT catch (using length of boat = 9 m, trip = 9 days at sea, SST = 29.25 °C, and chl *a* = 0.15 mg·m$^{-3}$) showed a fluctuating pattern for both monthly and annual CPUE (Figures 5 and 6). Monthly standardized CPUE ranged from 7.6 in February to 19.3 in October and the annual standardized CPUE ranged from 10.8 in 2017 to 21.2 in 2012.

When estimating the standardized CPUE from chl *a* concentration and SST, higher CPUEs were estimated at lower chl *a* concentrations, while CPUEs were not linearly influenced by SST (Figures 4 and 7).

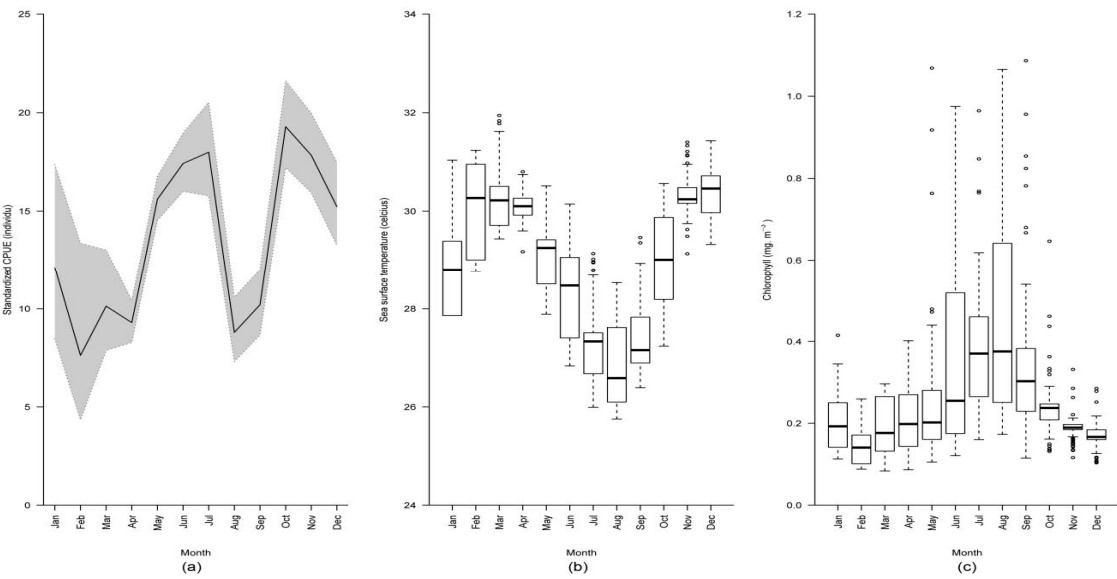

**Figure 5.** Seasonal (**a**) standardized catch per unit effort (CPUE) of large (>10 kg) Yellowfin Tuna *Thunnus albacares*, (**b**) seasonal SST, and (**c**) Chl *a* concentration from all fishing grounds plotted using boxplots.

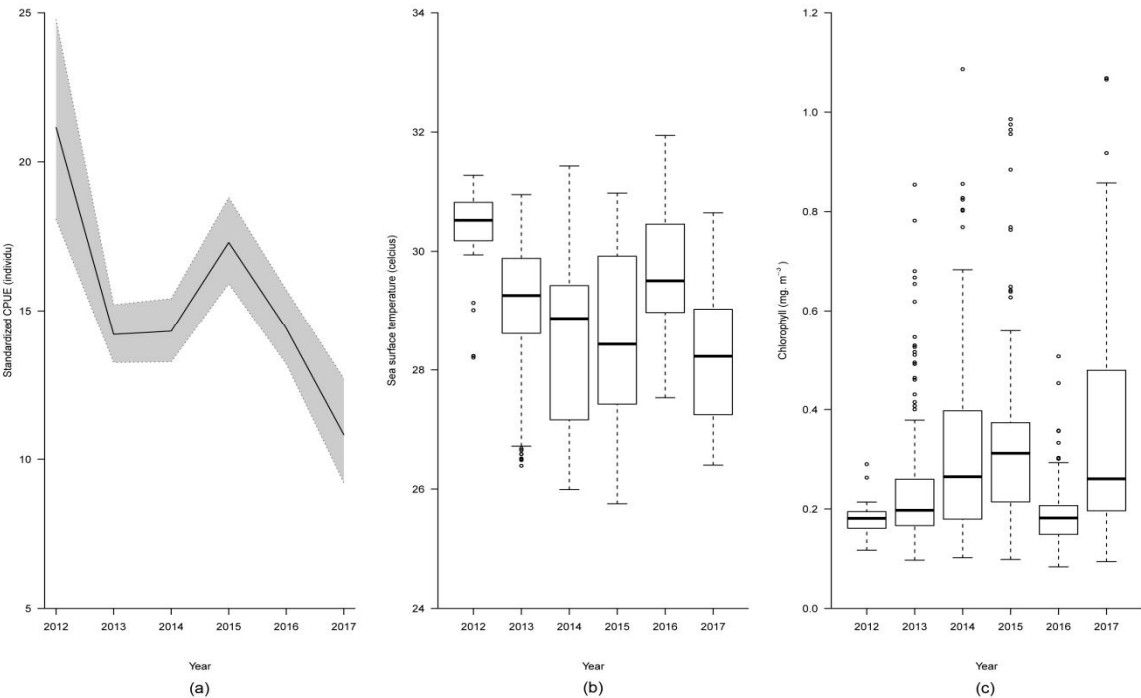

**Figure 6.** Annual (**a**) standardized CPUE of large (>10 kg) Yellowfin Tuna, *Thunnus albacares*, (**b**) monthly SST, and (**c**) Chl *a* concentration from all fishing grounds plotted using boxplots.

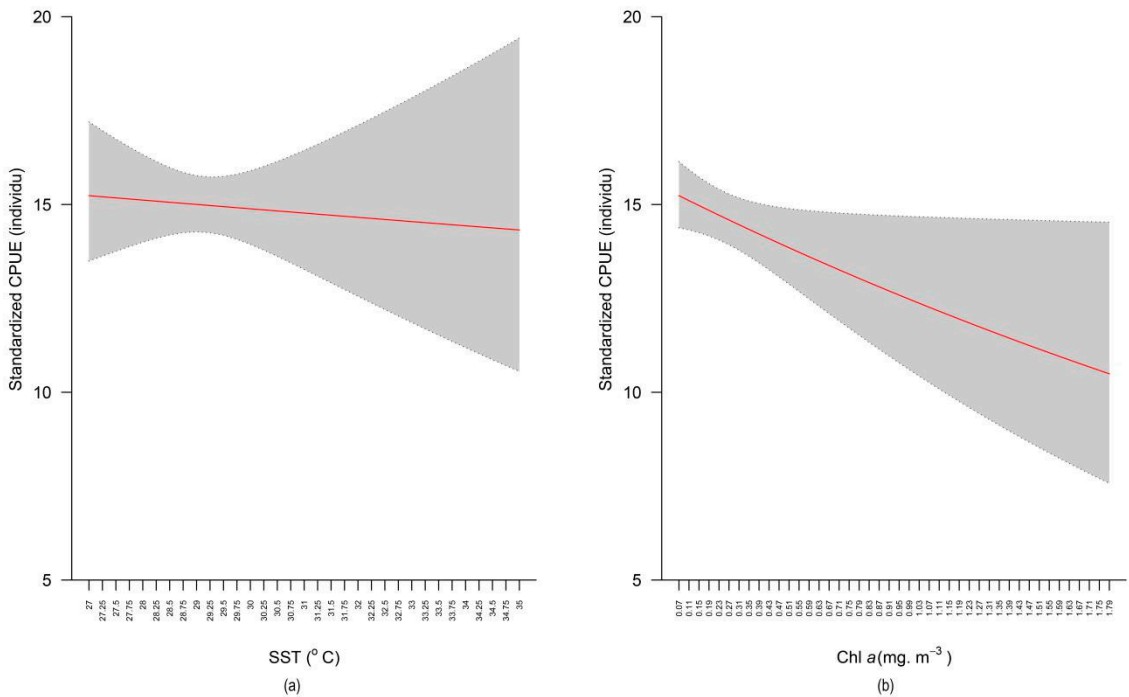

**Figure 7.** Standardized CPUE of large (>10 kg) Yellowfin Tuna, *Thunnus albacares* fitted using a General Additive Model by (**a**) variation of SST and (**b**) chl *a* concentration.

## 3. Discussion

Strong seasonal and annual patterns were recorded in the catch data of large (>10 kg) Yellowfin Tuna (YFT) in Lombok Island waters from 2012 to 2017. Based on GAM analyses, we found significant changes in the CPUE of large YFT recorded both among months of the year and among years. These patterns of YFT catch were also influenced by chl *a* concentration and ocean-atmosphere phenomena, such as sea-surface temperatures (SST) and El Niño events. The catch rates of large YFT were greater during the transition period of SST and chl *a* concentration, i.e., May–July (from rainy to dry seasons) and October–November (from dry to rainy seasons). During May and July, the SST decreased, and the chl *a* concentration increased, while the converse of this pattern was seen from October to November, i.e., SST increased and chl *a* concentration decreased. Chl *a* concentrations are influenced by SST [20], which in turn influences the net primary productivity of plankton. The abundance of tuna increases in high-productivity zones (such as upwelling zones, around islands, or at current interfaces), which attracts tuna to areas with greater food supply [21]. Therefore, the relative densities of the fish usually follow the distribution of primary production, with some time delays or lags.

Despite the limitations of the CPUE method, which include catch data and information about population dynamics that can help interpret productivity of fisheries and identify contradictions [22], our study overcome the shortcomings by adding environmental factors when considering the variation in standardized CPUE of large YFT among years. The greatest mean catches were recorded in 2012 and 2015, years with cooler monthly mean SSTs and mostly greater mean monthly chl *a* concentrations than in other years. A coupled ocean-atmosphere mode, namely an El Niño event (cooler waters and higher primary production), took place in the tropical Pacific Ocean during 2012 and 2015–2016, which might have influenced Lombok waters through the Indonesian Throughflow that passes through the Lombok Strait. In 2012, the El Niño occurred in August and September and was categorized as a weak El Niño [23], while in 2015, the El Niño event started to develop from April to May 2015, reached its maximum between November and December 2015, and finished in April–May 2016 [24]. This event had an El Niño index of 3.4 and was classified as a strong event [25], one of the strongest events recorded during the last two decades [24]. During El Niño events, it is likely that the water movement from Makassar Strait has lower flow [26], thus it could have influenced the upwelling level

in the coastal waters of Lombok and Sumbawa waters because cold water with higher productivity in deep water flows to the surface. During the El Niño events, the coastal upwelling level in terms of chl *a* concentration was more intense than in other years due to no disturbance from the Indonesia Throughflow waters from Lombok Strait [27]. The abundance of various economically important fish species such as tuna is influenced by environmental conditions and food availability [21,28]. To a certain degree, the water movement related to the El Niño event determines the distribution of water temperature [29]. The distribution and abundance of large YFT are closely associated with physical changes in the ocean, such as temperature changes, storms (length and magnitude or intensity), and the El Niño, as well as variations in the mixing depth layer and disturbances caused by the wind [21,30]. Our study emphasis on role of ENSO using longer time series data and investigating the spatial variability of key environmental variables SST and food distribution in the Indian Ocean in relation to tuna distribution and its relationship with climate on different time scales could further improve tuna habitat models, as also suggested by previous researchers [31].

According to the best selected GAM model, the standardized CPUE can be significantly predicted by specific factors, such as month, fishing trip (days at sea), year, boat size, and chl *a*. The SST and chl *a* concentrations are the main variables that contribute to the upwelling phenomenon that influences the abundance of fish in this area. The peak of chl *a* concentration occurred in July and August, which indicated annual upwelling that used to occur annually in this region from June (see Ningsih et al. [32]). This was followed by the high catch of the mature size of tuna in October, and slightly reduced catch in November–December during 2012–2017, about three months after the peak in chl *a*. The "time-lag" between the upwelling event and the season of tuna catch is due to effect of trophic conversion, where the tuna prey reaches the highest abundance 3–7 months after the peak of phytoplankton or pigment concentration [14,33,34]. The influence of chl *a* on the catch of tuna resulted in varying degrees of lag effect on trophic conversion. Ortega-Garcı́a and Lluch-Cota [34] found a 3- to 5-month time-lag between high chl *a* and high tuna abundance in the tropical Pacific, while Lehodey et al. [33] recorded a longer time-lag of 3 to 7 months. Meanwhile, the correlation between upwelling time may have a direct correlation with tuna juveniles only [35]. Although many studies concluded that SST has an influence on tuna abundance, we did not find a significant parametric effect of SST on the standardized CPUE of large YFT, only a non-linear effect. A plausible explanation for our finding is that SST contributed to a nonlinear effect on the abundance of YFT in Lombok Island waters. Dunstan et al. [20] stated that there is no consistent correlation and mechanism between SST and chl *a* variation, and warming water results in decreasing chl *a* concentration as well as leading to increased chl *a* concentration. Additionally, YFT can efficiently regulate their body temperature and allow themselves to make large horizontal and vertical migrations, hence reducing the effect of SST on the CPUE [14,36]. Lan et al. [14] found that subsurface water temperature was linearly affecting the abundance of YFT, rather than the SST itself.

This study revealed a significant correlation between environmental parameter fluctuation and regular seasonality and annual abundance of YFT, indicated by seasonal and annual patterns of variation in the standardized catch per unit of effort (CPUE) (Figure 5, Table 4). These dynamics of the large tuna influence the movements of tuna hand-line fishers, especially those who fish in grids without FADs. Currently, small-scale tuna management in West Nusa Tenggara has not yet adopted spatial management, e.g., using time/area closures on the tuna fishing grounds. These closures might be effective in conserving juvenile tuna and reducing bycatch. Our results suggest that the government could consider implementing a temporal closure in the waters of southern Lombok for two months after the chl *a* concentration reaches the highest level from August to September. At this time, the abundance of large YFT is low and the abundance of juvenile YFT is probably very high due to the upwelling phenomenon. A closure at this time would thus allow the juvenile tuna to grow without experiencing fishing mortality and could lead to greater recruitment of larger fish. A three-month closure on FAD fishing in the Atlantic appears to have been successful in reducing the catch of juvenile YFT and bigeye tuna [37]. The results of area and month closures for tuna fishing in the Banda Sea, of southern Maluku

Province, also indicate that a combination of area and temporal closures would aid in the recovery of YFT stocks and could potentially be beneficial for fisheries [38]. However, the decision of location choices, sizes, shapes, and duration of closure (seasonal or for a number of years) should be carefully studied and characterized by environmental parameters.

## 4. Materials and Methods

The waters of Indonesia's Exclusive Economic Zone (EEZ) are divided into 11 fisheries management areas (FMAs) that were formed in 1999 to optimize fisheries management in Indonesia [39]. The 11 FMAs fall into two major regional fishing areas recognised by the FAO: The Eastern Indian Ocean zone (FMAs 571, 572, and 573) and the Western and Central Pacific (FMAs 711, 712, 713, 714, 715, 716, 717, and 718). This study took place in Lombok Island, West Nusa Tenggara (WNT) Indonesia, part of FMA 713, which includes the relatively shallow waters of the Flores Sea and FMA 573 in the eastern Indian Ocean (Figure 1). The waters in this FMA are characterized as deep sea, with depths of up to 1000 m within 5 nautical miles of the coast in some locations. Upwelling events occur frequently in these waters due to its bathymetry, currents, and other oceanography features [32]. Small-scale fisheries predominate in this province and the most prominent landing sites in Lombok Island are Labuhan Lombok, Tanjung Luar, and various traditional landing sites, such as Bangko-bangko and Tanjung Ringgit (Figure 1a). Labuhan Lombok is the most important landing site for tuna on Lombok Island as this is where yellowfin tuna, skipjack tuna, and frigate tuna (*Auxis thazard*) are all landed at its coastal harbor (Figure 1).

### 4.1. Data Collection

#### 4.1.1. Fisheries Data

We used landing data for tuna from the Data Management Committee (DMC) of West Nusa Tenggara province, which oversees and manages data collection and management for the tuna fishery in the province. The provincial fisheries office employs three trained enumerators to collect data from each trip in Labuhan Lombok during working days (Monday to Friday; at least 20 days a month) and one supervisor to oversee and ensure data quality. We extracted data 2012 to 2017 from the Ifish system for this region and only used data from fishing trips that used handlines (Table 5), as this is the dominant fishing gear used in the tuna fishery in Lombok Island [18]. Note that data collection in 2012 started in February and in 2017 covered the months from January to October. The analyses concentrated on large yellowfin tuna (≥10 kg) that have reached maturity [10,40].

**Table 5.** Summary of the variables extracted from the Ifish data base provided through the data Management Committee of West Nusa Tenggara province in eastern Indonesia. Data retrieved December 2017.

| Variable | Type | Description |
|---|---|---|
| Year | Categorical | Year of trip (2012–2017) |
| Month | Categorical | Month of trip (January–December) |
| Days at sea | Numeric | Total day of fishing trip |
| Boat length | Numeric | Length of boat in meters |
| Fish aggregating device (FAD) | Categorical | Fishing on FAD or not |
| Fishing Ground | Categorical | Grid code of fishing areas (Figure 1) |
| Catch | Numeric | Number of fish with weight ≥10 kg caught on each fishing trip |

#### 4.1.2. Sea Surface Temperature and Chlorophyll Data

SST and chl *a* data were downloaded from the Environmental Research Division's Data Access Program (ERDDAP; https://coastwatch.pfeg.noaa.gov/erddap/info/index.html). SST measurements were gathered by the Moderate Resolution Imaging Spectroradiometer (MODIS) that was carried aboard by the NASA's Aqua spacecraft, while chl *a* measurements were also gathered using the MODIS

and generated according to algorithm developed by Hu et al., 2012 [41]. We selected the following data from this source: grid code (total area of a grid = 1.2 million ha) (see Figure 1b), days at sea for each fishing trip, and the catch of large YFT. We used average SST and chl *a* for each fishing trip: for trips of ≤3 d = average over the days of the trip; trips of 4–10 days in length = 8–day mean value; and for trips of >10 d at sea, the 30 d mean value. When the daily SST or chl *a* data were not available, we used moving averages (see Johnston et al. [42]) to estimate the missing values.

### 4.2. Data Analyses

Prior to the statistical analyses, we plotted all data (catch versus all factors) and then used analysis of variance [43] to understand the patterns of variation in unstandardized catch and the factors contributing towards this variation. Then, we used Generalized Linear Models (GLM) and Generalized Additive Models (GAMs) to standardize CPUE and to understand the influence of month of the year (as a proxy for the season), SST, and chl *a* on the abundance of large YFT. The full models for both types of analyses were:

1. CPUE~Year + month + boat length + days at sea + Chl *a* + SST … fitted with a simple linear regression or Generalized Linear Model (GLM);
2. CPUE~Year + month + *lo* (boat length) + *lo* (days at sea) + *lo* (Chl *a*) + *lo* (SST) … fitted with a Generalized Additive Model (GAM) using a loess/local regression, *lo* [44];
3. CPUE~Year + month + *s* (boat length) + *s* (days at sea) + *s* (Chl *a*) + *s*(SST) … fitted with a Generalized Additive Model (GAM) using a smoothing function (*s*(x)) [44].

All models were run using R software [45] and the best model was selected using the dredge function through automated model selection with subsets of the full model and using the Akaike information criterion (AIC) values [46]. We then inspected the residuals from selected models using GLMs with different families and GAMs with different smoothing functions, and compared the fit of the models using the AIC values to determine the most parsimonious model [47]. Multi-collinearity was checked using the variance inflation factor [48]. We estimated the standardized CPUE for each month, year, range of SST, and range of chl *a* according to the mode of the set of data values.

**Author Contributions:** Conceptualization, B.W. and I.Y.; methodology, B.W., I.Y., U.M., J.P., W., P.S.T. and D.D.; software, I.Y. and U.M.; validation, B.W., N.L., U.M., S.K., P.I.W., J.P., W., P.S.T., D.D. and I.Y.; formal analysis; B.W., N.L., U.M., S.K., P.I.W., J.P. and I.Y.; writing—original draft preparation, B.W. and I.Y.; writing—review and editing, B.W., N.L., U.M., S.K., P.I.W., J.P., W., P.S.T., D.D. and I.Y.; visualization, I.Y and U.M.; supervision, B.W., N.L., and I.Y. All authors have read and agreed to the published version of the manuscript.

**Funding:** This research received no external funding.

**Acknowledgments:** We are grateful to Lalu Wahyudi Adiguna and Nurjamil from Fisheries Co-Management Committee of West Nusa Tenggara Province and all staff of Masyarakat dan Perikanan Indonesia Foundation (MDPI) for supporting this research.

**Conflicts of Interest:** Deirdre Duggan, Wildan, and Putra Satria Timur are employee of Masyarakat dan Perikanan Indonesia Foundation (MDPI), who lead and organize the project on the Fisheries Improvement Program and Ifish data base for handline tuna fisheries in West Nusa Tenggara. Those data were used in this manuscript. However, we declare that MDPI affiliation do not influence the writing or interpretation of the results from the study reported in this manuscript. Other authors declare there is no conflict of interest related to this work.

## Appendix A

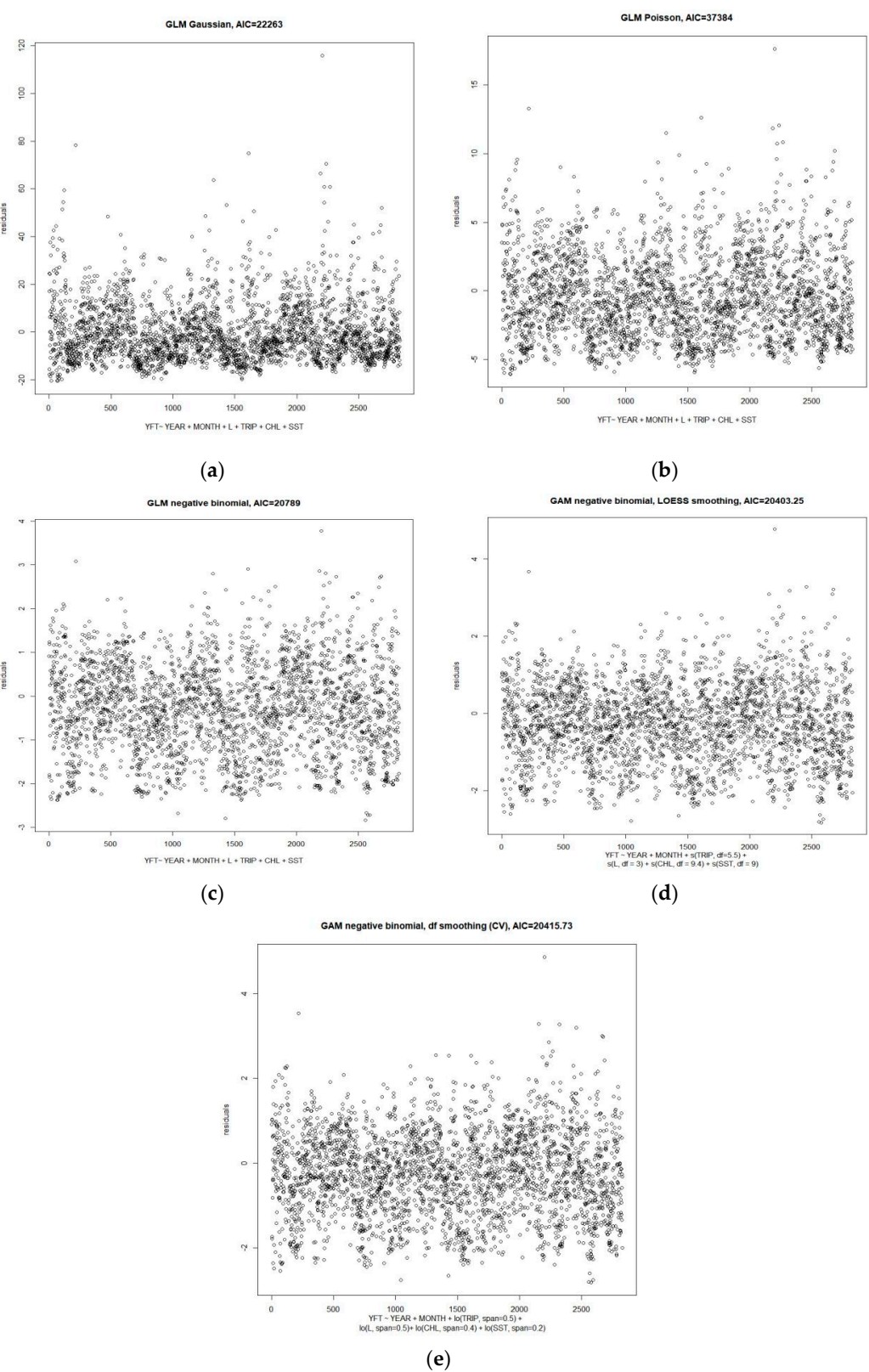

**Figure A1.** Residual plot and AIC of all full models tested using different GLM distributions (**a**–**c**) and different GAM smoothing lines (**d**,**e**).

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
