# Peer review of "Catch per Unit Effort Dynamic of Yellowfin Tuna Related to Sea Surface Temperature and Chlorophyll in Southern Indonesia"

_fishes, doi:10.3390/fishes5030028_

Round 1

Reviewer 1 Report

The authors have chosen an interesting and important topic. Please see my other comments and questions in the attachment.

Reviewer 2 Report

Dear Authors, 

This paper focuses on a timely topic: patterns of fishing intensities in Indonesia one of the most important regions in tuna exploitation. I consider this contribution a very well-done study that represents an important advance in understanding patterns in CPUE in relation to environmental features. In my opinion the manuscript is generally fine, though there are some areas where improvement is possible. 

Overall, the English is good, but there are places where the phrasing could be improved (i.e. language, spelling and formatting mistakes). I corrected the ones that I have observed (in track changes mode in the attached pdf). I am not a native speaker though. I would suggest to English proof-read the manuscript.

Please find major comments and suggestions bellow. Specific minor comments are detailed along the entire MS, attached to this page.

Abstract:

  • please state your aim and the importance of your study. Why it’s important to understand the identified patterns and to have an assessment of Catch per unit effort dynamic of yellowfin tuna in Indonesia? Is there  overexploitation and consequently patterns must be deducted to find the best preservation measures and best practices for sustainable fishing?

Introduction

  • please put your research in a wider context. There are several studies addressing the topic in different countries, even tough some are focused on different species. I think it’s important to give some insight into other studies focusing on the CPUE as robust indicator in describing impacts of fishing intensity in aquatic ecosystems/fisheries management (i.e. https://onlinelibrary.wiley.com/doi/abs/10.1111/lre.12061; https://www.tandfonline.com/doi/abs/10.1080/10641260600893766?scroll=top&needAccess=true&journalCode=brfs20). 

Results:

  • as presented now, the results are not easy to follow. I suggest to organise the results in subchapters: 1) first related to distribution patterns; 2) 2nd focused on patterns related to environmental features (the modelling results);
  • Lines 132-133: please give the exploratory ANOVA results in a table or give necessary details in the text (at least F statistic details);  
  • Figures should be given at a better resolution, with the text along the axes redrawn.

Discussion

  • please organise the Discussion by taking into account the recommendations from Results;
  • Discuss the limitations of the CPUE method (please see https://academic.oup.com/icesjms/article/63/8/1373/710477) and how your study overcomes the shortcomings;
  • please ad a final paragraph/Conclusion where you should list the recommendations for sustainable fishing in the area, as you mentioned at line 97 of your MS.

Materials and methods

  • are well described and explained. 

Stay safe and healthy! With my best regards.
